# Therapeutic Potential of BAY-117082, a Selective NLRP3 Inflammasome Inhibitor, on Metastatic Evolution in Human Oral Squamous Cell Carcinoma (OSCC)

**DOI:** 10.3390/cancers15102796

**Published:** 2023-05-17

**Authors:** Giovanna Casili, Sarah Adriana Scuderi, Marika Lanza, Alessia Filippone, Deborah Mannino, Raffaella Giuffrida, Cristina Colarossi, Marzia Mare, Anna Paola Capra, Federica De Gaetano, Marco Portelli, Angela Militi, Salvatore Cuzzocrea, Irene Paterniti, Emanuela Esposito

**Affiliations:** 1Department of Chemical, Biological, Pharmaceutical and Environmental Sciences, University of Messina, Viale Ferdinando Stagno D’Alcontres, 31, 98166 Messina, Italy; gcasili@unime.it (G.C.); sarahadriana.scuderi@unime.it (S.A.S.); mlanza@unime.it (M.L.); afilippone@unime.it (A.F.); deborah.mannino@unime.it (D.M.); annapaola.capra@unime.it (A.P.C.); fedegaetano@unime.it (F.D.G.); angela.militi@unime.it (A.M.); salvator@unime.it (S.C.); eesposito@unime.it (E.E.); 2IOM Ricerca, Via Penninazzo 11, 95029 Viagrande Catania, Italy; raffaella.giuffrida@grupposamed.com (R.G.); cristina.colarossi@grupposamed.com (C.C.); marzia.mare@grupposamed.com (M.M.); 3Department of Biomedical and Dental Science, Morphological and Functional Images, University of Messina, Via Consolare Valeria, 98125 Messina, Italy; marco.portelli@unime.it

**Keywords:** oral cancer, NLRP3, metastasis, lymph node, spleen, lung

## Abstract

**Simple Summary:**

Each year, new cases of oral cancer occur and metastasis represents the primary determinant for survival. Thus, there is a need to improve the preoperative assessment of metastatic risk. Scientific evidence discovered that BAY 11-7082, a powerful inhibitor of the NLRP3 inflammasome, is able to modulate cell invasion and migration and counteract the apoptosis process. The purpose of this study was to evaluate the effect of BAY-117082 in an in vivo orthotopic model of OSCC and its role in the invasiveness and metastasis processes in neighbor organs such as lymph node, lung, and spleen tissues.

**Abstract:**

Oral squamous cell carcinoma (OSCC) is a commonly occurring head and neck cancer and it is characterized by a high metastasis grade. The aim of this study was to evaluate for the first time the effect of BAY-117082, a selective NLRP3 inflammasome inhibitor, in an in vivo orthotopic model of OSCC and its role in the invasiveness and metastasis processes in neighbor organs such as lymph node, lung, and spleen tissues. Our results demonstrated that BAY-117082 treatment, at doses of 2.5 mg/kg and 5 mg/kg, was able to significantly reduce the presence of microscopic tumor islands and nuclear pleomorphism in tongue tissues and modulate the NLRP3 inflammasome pathway activation in tongue tissues, as well as in metastatic organs such as lung and spleen. Additionally, BAY-117082 treatment modulated the epithelial–mesenchymal transition (EMT) process in tongue tissue as well as in metastatic organs such as lymph node, lung, and spleen, also reducing the expression of matrix metalloproteinases (MMPs), particularly MMP2 and MMP9, markers of cell invasion and migration. In conclusion, the obtained data demonstrated that BAY-117082 at doses of 2.5 mg/kg and 5 mg/kg were able to reduce the tongue tumor area as well as the degree of metastasis in lymph node, lung, and spleen tissues through the NLRP3 inflammasome pathway inhibition.

## 1. Introduction

Oral squamous cell carcinoma (OSCC) represents a common type of head and neck cancer; it is characterized by poor prognosis [1,2]. Radical resection represents the better strategy for patients with moderate to advanced oral cancer [3]. The multidisciplinary therapy of surgical resection, radiation, and chemotherapy in patients with an advanced stage resulted in 20% of patients being identified to have developed distant metastasis (DM) [4]; although the frequency of DM decreased, DM reduces a patient’s quality of life and affects the clinical outcome. DM typically manifest itself in the lung (81.5%), followed by bone and liver (20%); only a few articles have reported a different localization of DM in the lymph nodes [5], spleen, kidney, and heart [6,7]. Despite the currently available therapeutic options for the treatment of oral cancer, the survival rate for patients with OSCC remains very low [8]. Oral cancer is characterized by a metastasis process which includes the detachment of cells from tumor tissue, and their invasion, proliferation, and evasion through the lymphatic or blood system [9]. Therefore, the identification of new therapeutic targets and new molecules capable of decreasing or preventing the progression of oral cancer represents a crucial purpose. In the field of cancer research, great interest has been dedicated to inflammasomes-mediated inflammation [10]. Inflammasomes are multi-protein complexes which consist of the nucleotide-binding and oligomerization domain (NOD)-like receptor (NLR), adapter protein apoptosis-associated speck-like protein-containing CARD (ASC), and caspase-1 [11]. The inflammasome complex can be activated by various stimuli which subsequently cleave pro-interleukine-1β (IL-1β) to its mature bioactive form via the activated caspase-1 [12]. The involvement of inflammasome in bladder cancer [13], gastric cancer [14], and leukemia [15] has been highlighted by several studies; however, the role of NLRP3 in DM due to OSCC has not been fully elucidated. Scientific evidence has shown that BAY 11-7082, a powerful inhibitor of the NLRP3 inflammasome [16], possesses various pharmacological abilities; it is also able to modulate the apoptosis process [17]. The purpose of this study was to evaluate the effect of BAY-117082, a selective NLRP3 inflammasome inhibitor, in an in vivo orthotopic model of OSCC and its role in the invasiveness and metastasis processes in neighbor organs such as lymph nodes, lung, and spleen tissues. 

## 2. Materials and Methods

### 2.1. Animals

BALB/c nude male mice were obtained from Jackson Laboratory (Bar Harbor, Hancock, ME, USA) and fed with a typical regimen and water ad libitum under pathogen-free conditions with a cycle of 12 h light/12 h dark. Animal study was accepted by the University of Messina (n 368/2019-PR released on 14 May 2019) according to Italian regulations for the use of animals. This study was approved by the University of Messina review board for the care of animals. All animal experiments were carried out in agreement with Italian (DM 116192) and European Union regulations (2010/63/EU amended by Regulation 2019/1010).

### 2.2. Cell Line

The OSCC cell line CAL27 was acquired from ATCC (Manassas, VA, USA). CAL27 cells were grown in Dulbecco’s modified Eagle’s medium (Invitrogen, Waltham, MO, USA) supplemented with 10% fetal bovine serum (FBS) (Invitrogen) and 100 U/mL penicillin and 100 μg/mL streptomycin (Sigma-Aldrich, St. Louis, MO, USA) at 37 °C with 5% CO_2_.

### 2.3. Experimental Design

The orthotopic model was performed as previously described [18,19]. Briefly, 1 × 10^6^ CAL27 cells in 20 μL of phosphate-buffered saline (PBS) were injected into the lateral portion of the tongues of animals using a sterile 0.5 mL insulin syringe. Mice in the control group were injected with the vehicle only. Then, the animals were randomly divided into 4 groups to receive the vehicle or BAY-117082 at the doses of 2.5 and 5 mg/kg every 3 days according to [16,17] and as previously described in our study [16]. BAY-117082 was dissolved in PBS with 0.001% of DMSO. After 30 days, the mice were sacrificed, and the tongue, lung, lymph nodes, and spleen were excised and processed to perform several analyses.

#### Experimental Groups

Sham group (vehicle): mice only received vehicle (PBS).

OSCC group: intraperitoneal (ip) administration of PBS after OSCC model induction.

OSCC+ BAY-117082 2.5 mg/kg: mice received BAY-117082 2.5 mg/kg dissolved in PBS by intraperitoneal administration.

OSCC++ BAY-117082 5 mg/kg: mice received BAY-117082 5 mg/kg dissolved in PBS by intraperitoneal administration.

### 2.4. Hematoxylin and Eosin (H&E) Staining

The H&E assay was performed as shown previously [20]. Briefly, samples from tongue tumors and metastases from lung, spleen, and lymph node were deparaffinized with xylene and stained with H&E staining. The images are shown at 10× magnification (100 μm of the Bar scale) using an Axiovision Zeiss microscope (Milan, Italy). The degree of metastasis was quantified in the lymph node, lung, and spleen to evaluate the metastasis foci, as described previously [21].

### 2.5. Immunohistochemical Localization of N-cadherin, E-cadherin, MMP-2, MMP-9, and NLRP3

Immunohistochemistry was conducted as explained previously [22]. Sections from tongue tumors and metastases from lung, spleen, and lymph node were incubated overnight at room temperature with different primary antibodies: anti-N-cadherin (sc-393933, 1:100; Santa Cruz Biotechnology, Dallas, TX, USA), anti-E-cadherin (sc-8426, 1:100; Santa Cruz Biotechnology, Dallas, TX, USA), anti-MMP2 (sc-13595, 1:100; Santa Cruz Biotechnology, Dallas, TX, USA), anti-MMP9 (sc-393859, 1:100; Santa Cruz Biotechnology, Dallas, TX, USA), and anti-NLRP3 (sc-34411, 1:500; Santa Cruz Biotechnology, Dallas, TX, USA). Later, the pieces were cleaned with PBS and incubated for 1 h with the secondary antibody (Santa Cruz Biotechnology, CA, USA). This was performed using a negative control with no primary antibody. For the immunohistochemistry, magnifications of 20× (50 µm scale bar) are displayed.

### 2.6. Western Blot Analysis

Western blot analyses on samples from tongue, lung, and spleen metastases was performed as described previously [22]. The membranes were incubated with primary antibodies: anti-NLRP3 (sc-34411, 1:500; Santa Cruz Biotechnology, Dallas, TX, USA), anti-ASC (sc-22514, 1:500; Santa Cruz Biotechnology, Dallas, TX, USA), anti-IL-1β (sc-32294, 1:500; Santa Cruz Biotechnology, Dallas, TX, USA), anti-IL-18 (sc-80051, 1:500; Santa Cruz Biotechnology, Dallas, TX, USA), anti-N-cadherin (sc-393933, 1:100; Santa Cruz Biotechnology, Dallas, TX, USA), anti-E-cadherin (sc-8426, 1:100; Santa Cruz Biotechnology, Dallas, TX, USA), anti-MMP2 (sc-13595, 1:100; Santa Cruz Biotechnology, Dallas, TX, USA), anti-MMP9 (sc-393859, 1:100; Santa Cruz Biotechnology, Dallas, TX, USA), and anti-βactin for cytosolic fraction (1:500; Santa Cruz Biotechnology; Dallas, TX, USA. sc-8432). Signals were perceived with an enhanced chemiluminescence (ECL) detection system mixture according to the manufacturer’s instructions (Thermo Fisher, Waltham, MA, USA).

### 2.7. Enzyme-Linked Immunosorbent Assay (ELISA) for NF-κB and IκBα 

The levels of NF-κBp65 and IκBα in tongue samples were measured by an ELISA kit according to the manufacturer’s instructions (NFκB-p65 ELISA Kit, Catalog No: E-EL-M0838; Elabscience; IκBα ELISA kit, Catalog No: MOES01330; AssayGenie). The homogenates were centrifuged for 5 min at 5000× *g*; then, the supernatants were collected and stored at −20 °C.

### 2.8. Materials

The reagents were obtained from Sigma-Aldrich (Milan, Italy). All stock solutions were made in PBS (Sigma-Aldrich, Milan, Italy).

### 2.9. Statistical Analysis

Data were analyzed with GraphPad Prism 7.04 software using one-way ANOVA analysis followed by a Bonferroni post hoc test for multiple comparisons. A *p*-value of less than 0.05 was considered significant. All values are indicated as a mean ± standard deviation (SD).

## 3. Results

### 3.1. BAY-117082 Treatment Reduced OSCC Growth

Histological analysis revealed that the OSCC group was characterized by the presence of microscopic tumor islands around the main tumor site, an archetypical feature of squamous cell carcinoma, irregular size, and nuclear pleomorphism compared to the sham group (Figure 1A,B). Nevertheless, BAY-117082 at doses of 2.5 mg/kg and 5 mg/kg significantly decreased the tumor area, restoring the tongue tissue architecture (Figure 1C,D). During the experiment, the OSCC group showed a significant reduction in body weight compared to the sham group; however, the BAY-117082 treatment at doses of 2.5 mg/kg and 5 mg/kg showed no significance (Figure 1E). 

### 3.2. BAY-117082 Treatment Reduced NLRP3 Inflammasome Pathway Activation in OSCC

The NLRP3 inflammasome pathway overactivation could contribute to oral cancer progression [23]. Therefore, we decided to evaluate the effect of BAY-117082 on the NLRP3 and ASC expression by Western blot analysis, demonstrating that the expression of NLRP3 and ASC are very high in the OSCC group compared to the sham group; nevertheless, BAY-117082 at both doses significantly decreased their expression in a dose-dependent manner (Figure 2A,B). When activated, the NLRP3 inflammasome stimulates the release of pro-inflammatory cytokines IL-1β and IL-18, which promote the progression of cancer [16]. According to this, we found an increase in the IL-1β and IL-18 expressions in the OSCC group; however, the treatment with BAY-117082 at doses of 2.5 mg/kg and 5 mg/kg significantly decreased their expression in a dose-dependent manner (Figure 2C,D).

### 3.3. BAY-117082 Treatment Modulated Epithelial–Mesenchymal Transition (EMT) and Matrix Metalloproteinases (MMPs) Expression in OSCC

Studies demonstrated that the epithelial–mesenchymal transition (EMT) plays a key role in the processes of oral cancer invasion and metastasis [24]. Thus, we decided to investigate the effect of BAY-117082 on N-cadherin and E-cadherin expression by immunohistochemical analysis, demonstrating that the OSCC group was characterized by an increase in N-cadherin and a decrease in E-cadherin expression compared to the sham groups (Figure 3A,B,F,G); however, the treatment with BAY-117082 at doses of 2.5 mg/kg and 5 mg/kg significantly reduced and increased the N-cadherin and the E-cadherin expressions, respectively, as shown in Figure 3C,D (see the % of the total tissue area score in Figure 3E) and Figure 3H,I (see the % of the total tissue area score in Figure 3J), respectively. Furthermore, we decided to evaluate the effect of BAY-117082 on the matrix metalloproteinase levels and particularly on MMP2 and MMP9, as markers of cell invasion and migration [25], showing that BAY-117082 at doses of 2.5 mg/kg and 5 mg/kg was able to significantly decrease their levels compared to the OSCC group, as shown in Figure 3K–N (see the % of the total tissue area score in Figure 3O); and in Figure 3P–S (see the % of the total tissue area score in Figure 3T).

### 3.4. BAY-117082 Treatment Reduced Metastasis Grade in OSCC Metastasis in Lymph Node, Lung, and Spleen

The metastasis process plays a pivotal role in oral cancer [26]. The most common site for OSCC metastasis is that of a cervical lymph node; however, tumor cells can move through lymphatic or blood vessels to distant metastatic sites such as the lung and the spleen [6]. Therefore, based on these considerations, we first decided to evaluate the ability of BAY-117082 to reduce the metastasis grade in the OSCC metastatic organs such as lymph nodes. Our results demonstrate that the OSCC group was characterized by the formation of an epidermoid cyst-like lesion lined by a layer of stratified squamous epithelium and nuclear pleomorphism compared to the sham group (Figure 4A,B,M); however, the treatment with BAY-117082 at doses of 2.5 mg/kg and 5 mg/kg was able to significantly decrease the lymphatic metastasis (Figure 4C,D,M). Additionally, we investigated whether BAY-117082 also reduced lung and spleen metastasis following OSCC induction. The histological analysis demonstrated that BAY-117082 at doses of 2.5 mg/kg and 5 mg/kg significantly reduced the diffuse tumor cell infiltration as well as the presence of multifocal cellular aggregates in lung tissues compared to the OSCC group (Figure 4E–H,N). Meanwhile, the histological analysis of spleen tissues showed that the tumor cells were spindled to polygonal with a poorly demarcated eosinophilic cytoplasm followed by the multinodular infiltration in the OSCC group compared to the sham group (Figure 4I,J); however, the treatment with BAY-117082 at both doses was able to significantly reduce the degree of metastasis (Figure 4K,L,O).

### 3.5. BAY-117082 Treatment Reduced NLRP3 Inflammasome Pathway Activation in OSCC Metastasis in Lung and Spleen

Considering the great selectivity and inhibitor effect of BAY-117082 against the NLRP3 inflammasome [27], we also decided to evaluate its effect in OSCC metastatic organs namely the lung and spleen tissues. In this context, our results demonstrated that the OSCC groups were characterized by an increase in NLRP3, ASC, IL-1β, and IL-18 expressions in the lung as well as in spleen tissues compared to sham groups, respectively; however, the treatment with BAY-117082 at doses of 2.5 mg/kg and 5 mg/kg was able to significantly decrease the NLRP3, ASC, IL-1β, and IL-18 expressions in both OSCC metastatic organs (see the lung panel in Figure 5A–D and the spleen panel in Figure 5E–H).

### 3.6. BAY-117082 Treatment Modulated Epithelial–Mesenchymal Transition (EMT) and Matrix Metalloproteinases (MMPs) Expression in OSCC Metastatic Lymph Node, Lung, and Spleen

Considering the role of EMT in oral cancer progression [24], we also decided to evaluate the effect of BAY-117082 on E-cadherin and N-cadherin expression in the lymph node, lung, and spleen tissues following OSCC induction. Our data demonstrated that the OSCC group was characterized by a decrease in E-cadherin expression in all three OSCC metastatic organs (as shown in Figure 6A,B,F,G,K,L); however, the treatment with BAY-117082 at doses of 2.5 mg/kg and 5 mg/kg significantly restored the E-cadherin expression, as can be seen in the lymph node in Figure 6C,D (see the % of the total tissue area score in Figure 6E); in the lung in Figure 6H,I (see the % of the total tissue area score in Figure 6J; and in the spleen in Figure 6M,N (see the % of the total tissue area score Figure 6O). Moreover, the immunohistochemical analysis showed that the OSCC groups of the lymph node, lung, and spleen tissues were characterized by a significant increase in N-cadherin expression compared to the sham groups (Figure 7A,B,F,G,K,L, respectively); however, BAY-117082 at both doses reduced its expression in all three OSCC metastatic organs, as shown in Figure 7C,D for the lymph node (see the % of the total tissue area score in Figure 7E); in Figure 7H,I for the lung (see the % of the total tissue area score in Figure 7J); and in Figure 7M,N for the spleen (see the % of the total tissue area score Figure 7O).

Additionally, we investigated the MMP levels by immunohistochemical analysis, demonstrating that the OSCC groups were characterized by a significant increase in MMP2 and MPP9 levels in the lymph node, lung, and spleen tissues compared to the sham groups (as shown in Figure 8A,B,F,G,K,L and Figure 9A,B,F,G,K,L, respectively); however, the treatment with BAY-117082 at both doses was able to significantly reduce their levels in all three OSCC metastatic organs, as shown in Figure 8C,D (see the % of the total tissue area score in Figure 8E); in Figure 8H,I (see the % of the total tissue area score in Figure 8J); in Figure 8M,N (see the % of the total tissue area score in Figure 8O); in Figure 9C,D (see the % of the total tissue area score in Figure 9E); in Figure 9H,I (see the % of the total tissue area score in Figure 9J); and in Figure 9M,N (see the % of the total tissue area score in Figure 9O). 

See Appendix A.

## 4. Discussion

OSCC is a malignant neoplasm derived from the stratified squamous epithelium of the oral mucosa; OSCC has an incidence of 450,000 new cases per year [28]. Smoking and excessive alcohol consumption represent the main risk factors for OSCC development; however, human papillomavirus (HPV), dietary deficiencies, and genomic modifications are also involved [29]. OSCC can provoke regional as well as distant metastasis (DM) [6]; DM represents the major problem for oral cancer; it is associated with advanced stages of oral tumors [30]. Interestingly, nodal metastasis appears when cancer cells at the primary site pass through lymphatic channels and migrate to cervical lymph nodes [6]. Usually, oral carcinomas spread from the primary tumor site to an anatomically distant site; however, tumor cells extravasate from the vessels into the stroma of the metastatic site, colonizing neighboring organs and forming macroscopic metastasis. It is notable that the lung is the most common site for distant metastasis in cases of head and neck OSCC [31]. However, metastasis to other organs, such as the spleen and liver, can also occur [32]. The treatment for oral cancer includes surgical resection, followed by chemotherapy and radiotherapy; despite the advances in therapy against OSCC, no significant decrease in mortality or potential effects against DM have been revealed; furthermore, DM worsens the prognosis and reduces the chances of successful treatment [6]. Although the pathophysiology of oral cancer remains unclear, in vivo and in vitro studies have revealed the involvement of NLRP3 inflammasome activation in contributing to the initiation and progression of oral cancer [5,23]; NLRP3 also plays a role in activating the invasion and metastasis, which seems to be tissue- and context-dependent [33]. Interestingly, proteolytic enzymes such as MMPs are involved in the metastasis process and their augmented production could be linked with the invasive and metastatic phenotype in various tumors [34]. In the process of metastasis in oral cancer, numerous MMPs have played a key role; studies have suggested that tumor stromal cells produced MMPs to promote tumor invasion [35]. Recently, BAY-117082, as a strong inhibitor of NLRP3 inflammasome, showed significant anti-tumor effects, suggesting its possible use as a promising treatment for oral cancer [16]. Therefore, based on the key roles of the NLRP3 inflammasome, in this study, the DM-promoting property of NLRP3 was demonstrated for the first time, evaluating the beneficial effect of BAY-117082 on reducing metastasis in an in vivo orthotopic model of OSCC. In our previous study, we highlighted the protective effects of BAY-117082 treatment in oral cancer [16]; in this study, the histological analysis on OSCC tongue samples confirmed the capacity of BAY-117082 to significantly reduce the tumor area, restoring the tongue tissue architecture. Evidence demonstrated that the NLRP3 inflammasome controls the innate immunity response through ASC activation, which subsequently activates the inflammatory response [36] by cleaving the cytokines pro-IL-1β and pro-IL-18 into their biologically active forms [37]. It is particularly interesting that BAY-117082 is known to prevent the organization of the ASC pyroptosome and NLRP3 inflammasome function through the alkylation of the cysteine residues of the NLRP3 ATPase region [27]. In this study, a significant reduction in NLRP3, ASC, IL-1β, and IL-18 expression was shown in the tumor’s samples from BAY-117082-treated mice compared to the OSCC group. The DM of oral cancer is a complex process involving the detachment of cells from the tumor tissue, the spread of cancer cells to tissues and organs beyond where the tumor originated, and the formation of secondary foci [38]; interestingly, OSCC can metastasize to the cervical lymph nodes and distant soft tissue metastases mostly occur in the lung [39]. Additionally, although DM incidence in the spleen is relatively low compared with that in other organs, cases have been recorded [40]. In this study, it was demonstrated for the first time that BAY-117082 treatment reduced DM in OSCC metastatic organs, namely the lymph node, lung, and spleen. The prodigious selectivity and inhibitor effect of BAY-117082 against the NLRP3 inflammasome was also demonstrated through the modulation of the NLRP3 pathway in OSCC metastatic organs, namely the lung and spleen tissues. The epithelial–mesenchymal transition (EMT) is a process in which epithelial cells change into mesenchymal cells, acquiring high mobility and an elevated migration grade, contributing to the progression of cancer [40]. In depth, it has been studied that, during EMT, epithelial cells undergo a suppression of genes responsible for the synthesis of components that form adherent junctions, such as E-cadherin, thereby causing the loss of cell adhesion and apical-basal polarity, followed by an increase in transcription factors associated with mesenchymal genes, such as N-cadherin, vimentin, fibronectin, and extracellular MMPs [41]. In this study, the treatment with BAY-117082 notably increased E-cadherin expression and significantly reduced the N-cadherin and MMP expression in the OSCC tumor samples; the same effects were also confirmed on the DM samples (for the lymph nodes, lung, and spleen), suggesting an innovative role of BAY-117082 as a modulator of the epithelial and mesenchymal transition process in oral cancer.

## 5. Conclusions

Therefore, BAY-117082 could represent an effective therapeutic strategy to reduce or counteract OSCC metastasis in lymph nodes, lung and spleen, thanks to its ability to modulate the NLRP3 inflammasome pathway and regulate the EMT process responsible for connecting the secondary metastatic tumors to the OSCC primary. Additional studies are necessary to better comprehend the role of these signaling pathways in metastases related to OSCC.

## Figures and Tables

**Figure 1 cancers-15-02796-f001:**
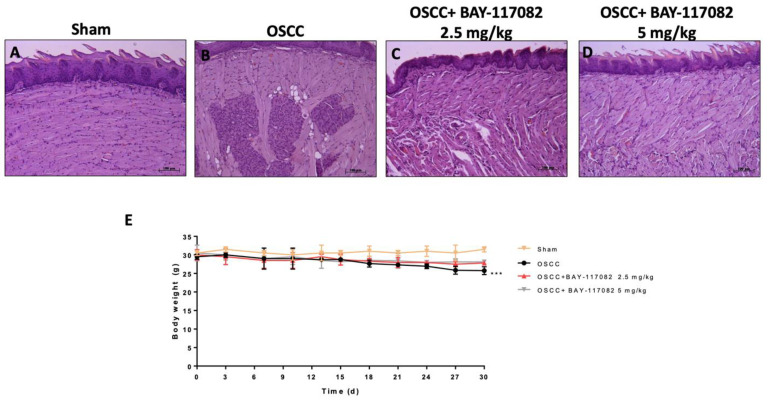
Effects of BAY-117082 treatment on tumor growth. The treatment with BAY-117082 2.5 mg/kg and 5 mg/kg significantly diminished the tumor mass (**C**,**D**) compared to the OSCC group (**B**). A significant decrease in animals’ body weights was observed in the OSCC group compared to sham group; however, BAY-117082 at both doses did not show any significance (**E**). Sections were observed and photographed at 10× magnification. (**E**) *** *p* < 0.01 vs. sham group.

**Figure 2 cancers-15-02796-f002:**
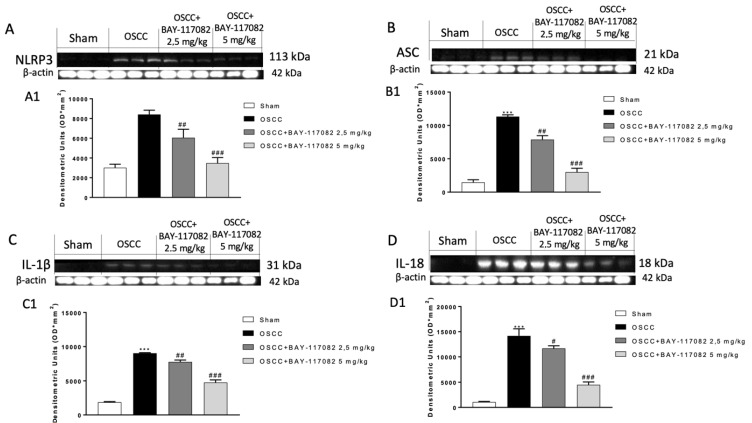
BAY-117082 reduced the NLRP3 inflammasome pathway activation in tongue samples. Western blot analysis showed that BAY-117082 2.5 mg/kg and 5 mg/kg decreased NLRP3, ASC, IL-1β, and IL-18 expression compared to the OSCC group. Data are representative of at least three independent experiments. (**A**–**C**) *** *p* < 0.001 vs. sham; ## *p* < 0.01 and ### *p* < 0.001 vs. OSCC; (**D**) *** *p* < 0.001 vs. control; # *p* < 0.05 and ### *p* < 0.001 vs. OSCC. The uncropped bolts are shown in Appendix A.

**Figure 3 cancers-15-02796-f003:**
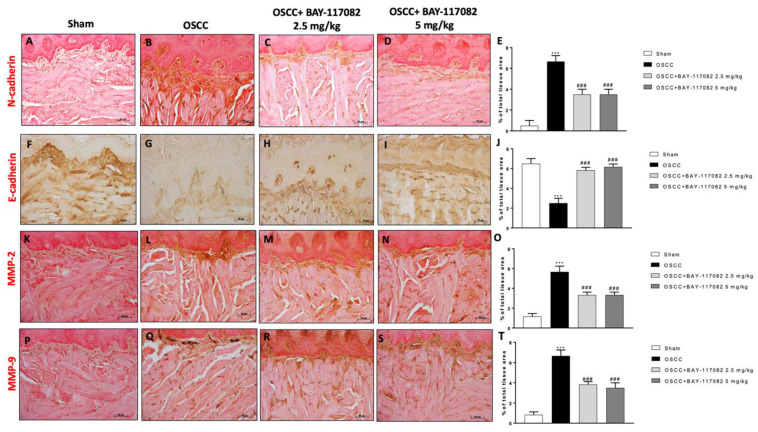
BAY-117082 treatment modulated the EMT process and MMP expression in tongue samples. Immunohistochemical analysis revealed that the treatment with BAY-117082 at doses of 2.5 and 5 mg/kg was able to reduce N-cadherin (**C**,**D**,**M**,**N**,**R**,**S**), MMP2, and MMP9 staining compared to the OSCC group (**B**,**L**,**Q**), increasing E-cadherin (H and I). Data are representative of at least three independent experiments. (**A**–**D**) *** *p* < 0.001 vs. sham; ### *p* < 0.001 vs. OSCC. (Scale bar: 50 μm).

**Figure 4 cancers-15-02796-f004:**
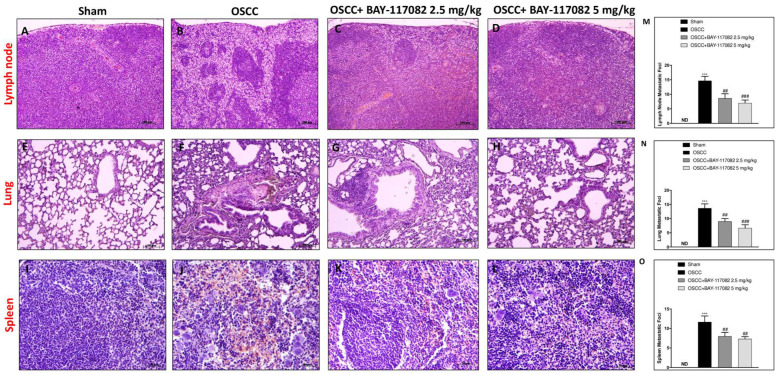
BAY-117082 treatment modulated the metastasis process in the lymph node, lung, and spleen. Histological analysis revealed that the treatment with BAY-117082 at doses of 2.5 and 5 mg/kg was able to reduce the degree of metastasis in the lymph node (**C**,**D**,**M**), lung (**G**,**H**,**N**), and spleen (**K**,**L**,**O**) in the C group compared with the OSCC group (**B**,**F**,**J**). (**A**,**E**,**I**) control groups in the lymph node, lung, and spleen. Data are representative of at least three independent experiments. ND, not designed. *** *p* < 0.001 vs. sham; ### *p* < 0.001 and vs. ## *p* < 0.01 OSCC.

**Figure 5 cancers-15-02796-f005:**
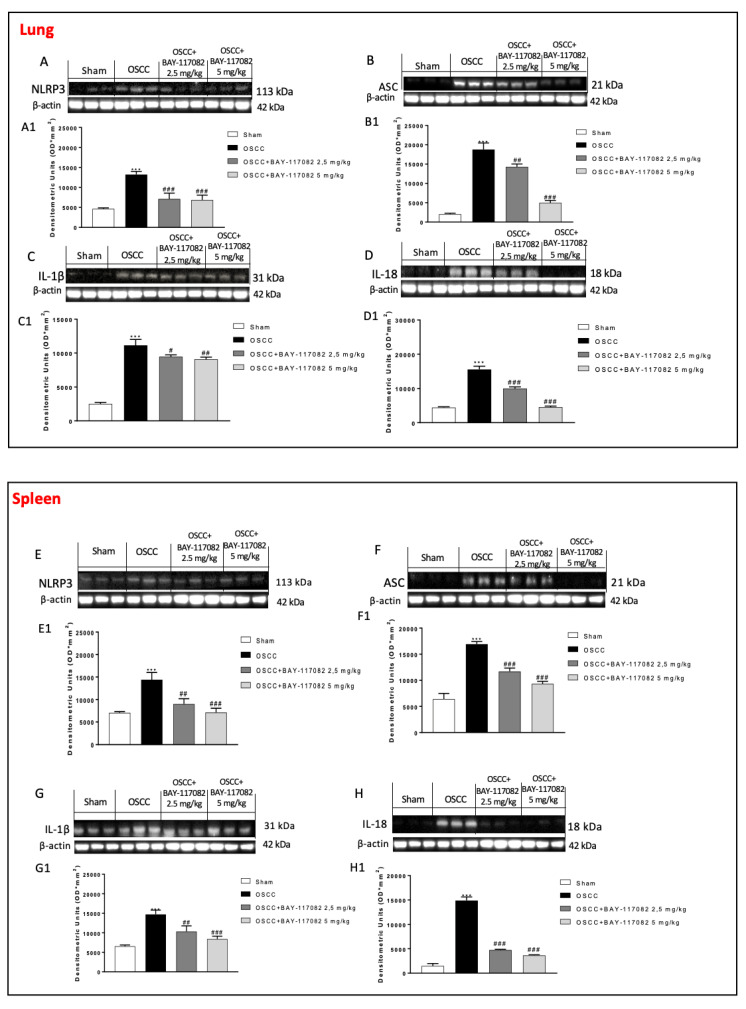
Effect of the BAY-117082 on the NLRP3 inflammasome pathway in lung and spleen samples. The blots revealed that the treatment with BAY-117082 at doses of 2.5 and 5 mg/kg was able to reduce the NLRP3, ASC, IL-1β, and IL-18 expressions compared to the OSCC group, in both the lung (**A**–**D**) and spleen (**E**–**H**). Data are representative of at least three independent experiments. (**A**,**D**,**F**,**H**) *** *p* < 0.001 vs. sham and ### *p* < 0.001 vs. OSCC; (**B**,**E**,**G**) *** *p* < 0.001 vs. sham; ## *p* < 0.01 and ### *p* < 0.001 vs. OSCC; and (**C**) *** *p* < 0.001 vs. sham; # *p* < 0.05 and ## *p* < 0.01 vs. OSCC.

**Figure 6 cancers-15-02796-f006:**
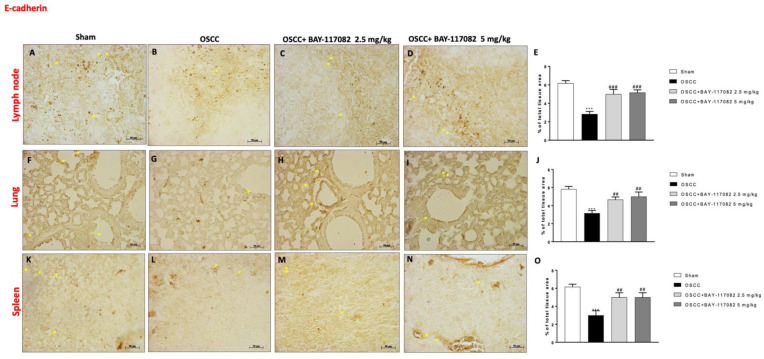
BAY-117082 treatment modulated the EMT pathway in the lymph node, lung, and spleen. Immunohistochemical analysis revealed that treatment with BAY-117082 at doses of 2.5 and 5 mg/kg was able to increase E-cadherin in the lymph node (**C**,**D**), lung (**H**,**I**), and spleen (**M**,**N**) compared to the OSCC group (**B**,**G**,**L**). Data are representative of at least three independent experiments. (**A**–**D**) *** *p* < 0.001 vs. sham; ## *p* < 0.01 and ### *p* < 0.001 vs. OSCC. (Scale bar: 50 μm).

**Figure 7 cancers-15-02796-f007:**
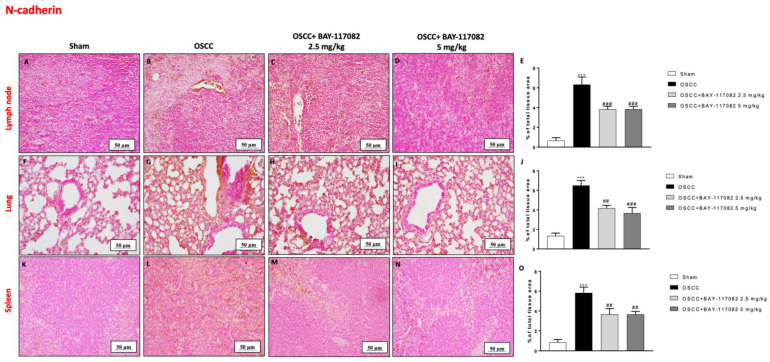
BAY-117082 treatment modulated the EMT pathway in the lymph node, lung, and spleen. Immunohistochemical analysis revealed that treatment with BAY-117082 at doses of 2.5 and 5 mg/kg was able to reduce N-cadherin in the lymph node (**C**,**D**), lung (**H**,**I**), and spleen (**M**,**N**) compared to the OSCC group (**B**,**G**,**L**). Data are representative of at least three independent experiments. (**A**–**D**) *** *p* < 0.001 vs. sham; ## *p* < 0.01 and ### *p* < 0.001 vs. OSCC.

**Figure 8 cancers-15-02796-f008:**
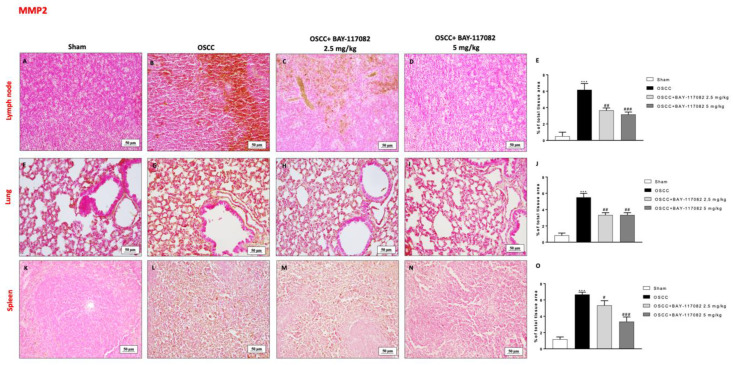
BAY-117082 treatment modulated MMP2 in the lymph node, lung, and spleen. Immunohistochemical analysis revealed that treatment with BAY-117082 at doses of 2.5 and 5 mg/kg was able to reduce MMP2 in the lymph node (**C**,**D**), lung (**H**,**I**), and spleen (**M**,**N**) compared to the OSCC group (**B**,**G**,**L**). Data are representative of at least three independent experiments. (**A**–**D**) *** *p* < 0.001 vs. sham; # *p* < 0.05, ## *p* < 0.01 and ### *p* < 0.001 vs. OSCC.

**Figure 9 cancers-15-02796-f009:**
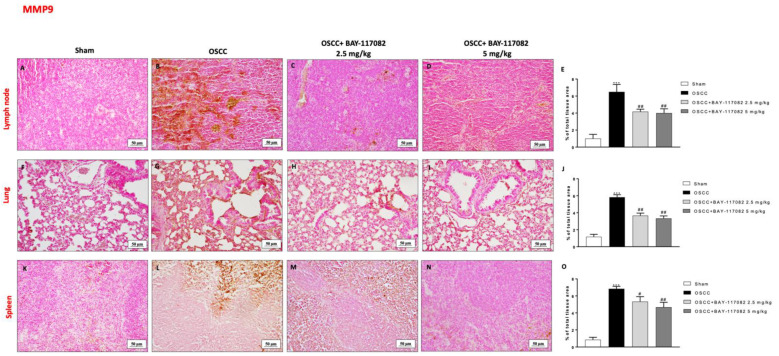
BAY-117082 treatment modulated MMP9 in the lymph node, lung, and spleen. Immunohistochemical analysis revealed that treatment with BAY-117082 at doses of 2.5 and 5 mg/kg was able to reduce MMP9 in the lymph node (**C**–**E**), lung (**H**–**J**), and spleen (**M**–**O**) compared to the OSCC group (**B**,**G**,**L**). Data are representative of at least three independent experiments. Control group (**A**,**F**,**K**)) *** *p* < 0.001 vs. sham; # *p* < 0.05 and ## *p* < 0.01 vs. OSCC.

## Data Availability

The data can be shared up on request.

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
