# Peer review of "Therapeutic Potential of BAY-117082, a Selective NLRP3 Inflammasome Inhibitor, on Metastatic Evolution in Human Oral Squamous Cell Carcinoma (OSCC)"

_cancers, 2023, doi:10.3390/cancers15102796_

Round 1

Reviewer 1 Report

Although the manuscript entitled “NLRP3 as a biomarker of metastatic evolution in human oral squamous cell carcinoma (OSCC)” is well written it has some shortcomings that I wrote below: 

-Since BAY-117082 is an inhibitor of both NF-κB and NLRP3 inflammasome pathways, the authors should also determine the expressions of NF-κB pathway members. The regression in OSCC might be because of NF-κB inhibition also.   

-The authors wrote that “All stock solutions were made in non-pyrogenic saline” in the methods section but they wrote sham group (vehicle) mice received only vehicle (PBS). If all solutions were made in saline why did the authors use PBS for the sham group?

-What dye did the authors use for counterstain in immunohistochemistry? They used this dye for the immunohistochemistry of Figures 3, 7, 8, and 9 but they did not use it for the immunohistochemistry of Figure 6. In Figures 7, 8, and 9 it is hard to see the immunoreactions. The authors should reduce the use of counterstain and add new photos.

-The authors wrote that BAY-117082 (2.5 and 5 mg/kg) decreased metastasis degree in the lymph node, lung, and spleen compared to the OSCC group. Did they measure the metastasis degree?

-The letters in Figure 4 are hard to see. It will be better to write them with dark colors. 

-If BAY-117082 is an NLRP3 inhibitor, how do the authors explain the decrease of ASC levels, as shown by the authors in Figure 2B, Figure 5B-B1, and F-F1. They should discuss it in the Discussion section.

-The authors did only Western blotting for NLRP3 inflammasome pathway members, they should also do immunohistochemistry. In a similar way they did immunohistochemistry for N-cadherin, E-cadherin, MMP-2, and MMP-9 and they should also do Western blotting for these markers in all organs they used. It will be more accurate to comment on the results.

Author Response

Reviewer 1

Although the manuscript entitled “NLRP3 as a biomarker of metastatic evolution in human oral squamous cell carcinoma (OSCC)” is well written it has some shortcomings that I wrote below:

-Since BAY-117082 is an inhibitor of both NF-κB and NLRP3 inflammasome pathways, the authors should also determine the expressions of NF-κB pathway members. The regression in OSCC might be because of NF-κB inhibition also.  

As suggested by reviewer, the authors investigated NF-κBp65 and IκBα protein expression in tongue samples using ELISA kits. The results demonstrated also the involvement of NF-κB inhibition in the regression in OSCC, as showed in Figure Supplementary 1.

-The authors wrote that “All stock solutions were made in non-pyrogenic saline” in the methods section but they wrote sham group (vehicle) mice received only vehicle (PBS). If all solutions were made in saline why did the authors use PBS for the sham group?

As suggested by reviewer, the authors corrected the sentence in 2.8 section Materials, specifying the use of PBS as vehicle for sham group and for BAY-117082 preparations.

-What dye did the authors use for counterstain in immunohistochemistry? They used this dye for the immunohistochemistry of Figures 3, 7, 8, and 9 but they did not use it for the immunohistochemistry of Figure 6. In Figures 7, 8, and 9 it is hard to see the immunoreactions. The authors should reduce the use of counterstain and add new photos.

The authors performed immunohistochemistry using Nuclear fast red for counterstain in all Figures, except for Figure 6 because this analysis has been performed later after reviewer suggestion. As suggested by reviewer, the authors reduced the counterstain and added new photos, as observed in the new Figures 7, 8 and 9.

-The authors wrote that BAY-117082 (2.5 and 5 mg/kg) decreased metastasis degree in the lymph node, lung, and spleen compared to the OSCC group. Did they measure the metastasis degree?

The authors measured metastasis degree through the evaluation of metastasis foci, respectively in lymph node, lung and spleen, as observed in the revised Figure 4.

-The letters in Figure 4 are hard to see. It will be better to write them with dark colors.

As suggested by reviewer, the letters in revised Figure 4 have been written better with dark color.

-If BAY-117082 is an NLRP3 inhibitor, how do the authors explain the decrease of ASC levels, as shown by the authors in Figure 2B, Figure 5B-B1, and F-F1. They should discuss it in the Discussion section.

Various scientific studies highlighted as NLRP3 inflammasome activation, by NLRP3 stimuli, triggers formation of a large ASC oligomer termed the ASC pyroptosome, which is composed of oligomerized ASC dimers. Therefore, upon activation, NLRP3 protein recruits the adapter ASC protein, which recruits the procaspase-1 resulting in its cleavage and activation, inducing the maturation and secretion of inflammatory cytokines and pyroptosis. (“Anti-inflammatory Compounds Parthenolide and Bay 11-7082 Are Direct Inhibitors of the Inflammasome” by Juliana C et al., 2010; “Pharmacological Inhibitors of the NLRP3 Inflammasome” by Zahid A et al., 2019). Interestingly, it is known that BAY-117082 prevents the organization of ASC pyroptosome and NLRP3 inflammasome function through alkylation of cysteine residues of NLRP3 ATPase region (“Anti-inflammatory Compounds Parthenolide and Bay 11-7082 Are Direct Inhibitors of the Inflammasome” by Juliana C et al., 2010); this explains the decrease of ASC levels following NLRP3 inhibition by BAY-117082. As suggested, the authors argued it in Discussion section.

- The authors did only Western blotting for NLRP3 inflammasome pathway members, they should also do immunohistochemistry. In a similar way they did immunohistochemistry for N-cadherin, E-cadherin, MMP-2, and MMP-9 and they should also do Western blotting for these markers in all organs they used. It will be more accurate to comment on the results.

As suggested by reviewer, the authors performed NLRP3 immunohistochemistry on tongue, lymph node, liver and spleen samples, as observed in Figure Supplementary 2. Moreover, in the similar way, the authors performed Western blot analysis for detect N-cadherin, E-cadherin, MMP-2 and MMP-9 in tongue, lymph node, liver and spleen samples, as showed in Supplementary Figures 3 and 4.

Reviewer 2 Report

I read with interest the paper by Dr Casili and his colleagues on NLRP3 as biomarker of metastatic evolution in human oral squamous cell carcinoma (OSCC).

This paper presents an interest and answers an important question in this area.

The results in the body of the manuscript are clearly described and the pictures are representative of the experiments carried out.

Author Response

Reviewer 2

I read with interest the paper by Dr Casili and his colleagues on NLRP3 as biomarker of metastatic evolution in human oral squamous cell carcinoma (OSCC).

This paper presents an interest and answers an important question in this area.

The results in the body of the manuscript are clearly described and the pictures are representative of the experiments carried out.

The authors thank the reviewer for their appreciation of the study.

Reviewer 3 Report

Overall:

The study investigated the potential therapeutic effect of BAY-117082, a selective NLRP3 inflammasome inhibitor, on an in vivo orthotopic model of oral squamous cell carcinoma (OSCC), as well as its role in reducing invasiveness and metastasis in organs such as the lymph node, lung, and spleen tissues. The findings demonstrated that treatment with BAY-117082 led to a significant reduction in tumor area and metastasis degree in these organs by inhibiting the NLRP3 inflammasome pathway. Moreover, the study sheds light on the crucial role of inflammasome-mediated inflammation in the development of OSCC.

Comment:

  1. The abstract provides a clear and concise overview of the study's findings, which could be helpful for readers interested in the field of OSCC and the role of inflammasomes in cancer biology. However, I would recommend that the authors revise the title to better reflect the primary objective of their study, which was to evaluate the therapeutic potential of BAY-117082 in OSCC. Although the title does mention NLRP3 as a biomarker of metastatic evolution in OSCC, it fails to capture the novelty and significance of your study, which is the evaluation of BAY-117082 as a potential treatment for OSCC. Therefore, I suggest that the title be revised to better reflect the main objective of your study. A more appropriate title could be "Therapeutic potential of BAY-117082, a selective NLRP3 inflammasome inhibitor……” This revised title more accurately conveys the focus of your study and highlights the potential clinical implications of your findings.
  2. It would be beneficial if the authors could provide a graphic representation of the mechanism through which BAY-117082 inhibits the NLRP3 inflammasome pathway. This visual aid could help readers better understand the study's findings and could be a valuable addition to the manuscript.

Author Response

Reviewer 3

Overall:

The study investigated the potential therapeutic effect of BAY-117082, a selective NLRP3 inflammasome inhibitor, on an in vivo orthotopic model of oral squamous cell carcinoma (OSCC), as well as its role in reducing invasiveness and metastasis in organs such as the lymph node, lung, and spleen tissues. The findings demonstrated that treatment with BAY-117082 led to a significant reduction in tumor area and metastasis degree in these organs by inhibiting the NLRP3 inflammasome pathway. Moreover, the study sheds light on the crucial role of inflammasome-mediated inflammation in the development of OSCC.

Comment:

The abstract provides a clear and concise overview of the study's findings, which could be helpful for readers interested in the field of OSCC and the role of inflammasomes in cancer biology. However, I would recommend that the authors revise the title to better reflect the primary objective of their study, which was to evaluate the therapeutic potential of BAY-117082 in OSCC. Although the title does mention NLRP3 as a biomarker of metastatic evolution in OSCC, it fails to capture the novelty and significance of your study, which is the evaluation of BAY-117082 as a potential treatment for OSCC. Therefore, I suggest that the title be revised to better reflect the main objective of your study. A more appropriate title could be "Therapeutic potential of BAY-117082, a selective NLRP3 inflammasome inhibitor……” This revised title more accurately conveys the focus of your study and highlights the potential clinical implications of your findings.

As suggested by reviewer, the authors revised the title with another more appropriate, to better conveys the focus of this study and to highlight the potential clinical implications of findings.

It would be beneficial if the authors could provide a graphic representation of the mechanism through which BAY-117082 inhibits the NLRP3 inflammasome pathway. This visual aid could help readers better understand the study's findings and could be a valuable addition to the manuscript.

As suggested by reviewer, the authors provided a graphical abstract to elucidate the mechanism through which BAY-117082 inhibits NLRP3 inflammasome pathway.

Round 2

Reviewer 1 Report

The authors did revisions suggested by the reviewers. The manuscript can be published in this form.